

# Few years' experience with Automatic DIFlux systems: theory, validation and results.

Poncelet Antoine[1] , Gonsette Alexandre[1] , and Rasson Jean[1]

1Royal Meteorological Institut of Belgium, rue du centre de hysique 4 , 5670 Dourbes

Correspondence to: Poncelet (antoine.poncelet@meteo.be)

**Abstract.** The previous release of our Automatic DIFlux, called AutoDIF mk2.2, is now running continuously since June 2012 in the absolute house of Dourbes magnetic observatory performing measurement every 30minutes. A second one is working in the tunnel of Conrad observatory (Austria) since December 2013. After this proof of concept, we improved the

AutoDIF to the version mk2.3 which was presented in 16[th] IAGA workshop in Hyderabad. Today, we have successfully deployed 6 AutoDIFs in various environments: 2 in Dourbes (DOU), 1 in Manhay (MAB), 1 in Conrad (CON), 1 in Deajeon (Korea) and 1 is used for tests. This one was installed for 10 month in Chambon-la-Foret (CLF) and since 2016 in Kakioka (KAK). In this paper, we will compare the automatic measurements with the human-made, and discuss the advantages / disadvantages of automatic measurements

## 1 Introduction


After some years of development the release of our Automatic DIFlux (Van Loo et al. , 2007)(Rasson et al. , 2011), called AutoDIF mk2.2, is now running continuously since June 2012 in the absolute house of Dourbes observatory (DOU). A second one is aslo running in Conrad Observatory (CON) since 2013. These two long time series give us feedback on the strengths and weaknesses of these instruments. This paper will first present the calibration or validation of the most

important sensors (fluxgate, level and encoders) and then, the baseline results of AutoDIFmk2.2 for observatories of Dourbes and Conrad. Based on these results and our experience, we will give the improvement we have made to achieve the AutoDIFmk2.3. Then, we will present some baseline results for this last version of AutoDIF.

## 2 Theory of Automated DIFlux

Lauridsen and Kerridge had established a DIFlux model giving the magnetic sensor output value according to its orientation

in space. On top of the two degrees of freedom (DOF) related to the rotation axes, sensor offset and misalignment errors were taken into account leading to a system with 5 DOF. We propose here to extend those models in order to include a possible levelling error consisting of two angular DOF, one toward geographic north and the other toward east. The vector


magnetic field is then expressed in the sensor frame. The first term of Eq. (1) gives the magnetic sensor output according to its 7 DOF.

$$\vec{T} = \begin{bmatrix} 1 & -\epsilon_f & 0 \\ \epsilon_f & 1 & -\gamma_f \\ 0 & \gamma_f & 1 \end{bmatrix} R_y(\beta) R_z(\phi) \begin{bmatrix} 1 & 0 & -A \\ 0 & 1 & -B \\ A & B & 1 \end{bmatrix} \begin{bmatrix} X \\ Y \\ Z \end{bmatrix} + \begin{bmatrix} T_{0x} \\ T_{0y} \\ T_{0z} \end{bmatrix},$$

(1)

Where:

- $A$      Tilt angle in the geographic north direction
- $B$      Tilt angle in the east direction
- $T_0$      Sensor offset
- $\phi$      Rotation angle around vertical axis
- $\beta$      Rotation angle around horizontal axis

- $\epsilon$      Collimation error in a vertical plan
- $\gamma$      Collimation error in a horizontal plan
- $\vec{T}$      Magnetic field vector in the sensor frame

Considering the following transformation: $\begin{bmatrix} X \\ Y \end{bmatrix} = \begin{bmatrix} H \cos(D) \\ H \sin(D) \end{bmatrix}$ the development of the first term of Eq. (1) leads to the

AutoDIF model (Eq. 2) .

$$T_{fl} = H\cos(\phi - D)(\cos(\beta) - \epsilon \sin(\beta)) - \gamma H \sin(\phi - D) - Z(\sin(\beta) - \epsilon \cos(\beta)) - \cos(\beta) Z(A\cos(\phi) +$$
$$B\sin(\phi)) - \sin(\beta) H(A\cos(D) + B\sin(D)) + T_{0x},$$

(2)

**2.1 Declination**

Declination measurement is performed by putting magnetic sensor perpendicular to the field in a horizontal plan. Therefore,

4configurations are possible: $\beta = 0, \pi$ and $= \frac{\pi}{2}, \frac{3\pi}{2}$. The 4 configurations are commonly designed according to the sensor pointing direction and its position relative to the horizontal axis. We thus have East-UP, West-Down, East-Down and West-UP. Keeping small angles approximations and removing second order terms such as $\epsilon A$ or $\epsilon \beta_0$:

$$T_{EU} \approx H(\phi - D) + \frac{\pi}{2} - \gamma H - Z\left(\beta_{EU} - \epsilon + (A\cos(\phi) + B\sin(\phi))\right) + T_{0x},$$

(3)

$$T_{WD} \approx -H(\phi - D) - \frac{\pi}{2} - \gamma H - Z\left(\beta_{WD} + \epsilon - (A\cos(\phi) + B\sin(\phi))\right) + T_{0x},$$

(4)

$$T_{ED} \approx H(\phi - D) + \frac{\pi}{2} + \gamma H - Z\left(\beta_{ED} + \epsilon + (A\cos(\phi) + B\sin(\phi))\right) + T_{0x},$$

(5)

$$T_{WU} \approx -H(\phi - D) - \frac{\pi}{2} + \gamma H - Z\left(\beta_{WU} - \epsilon - (A\cos(\phi) + B\sin(\phi))\right) + T_{0x},$$

(6)





Or, if $\phi$ is isolated:

$$\phi_{EU} \approx \frac{T_{EU}-T_{0x}}{H} + D + \frac{\pi}{2} + \gamma + \tan(I)\left(\beta_{EU} - \epsilon + (A\cos(\phi) + B\sin(\phi))\right), \tag{7}$$

$$\phi_{WD} \approx -\frac{T_{WD}-T_{0x}}{H} + D + \frac{\pi}{2} - \gamma - \tan(I)\left(\beta_{WD} + \epsilon - (A\cos(\phi) + B\sin(\phi))\right), \tag{8}$$

$$\phi_{ED} \approx \frac{T_{ED}-T_{0x}}{H} + D + \frac{\pi}{2} - \gamma + \tan(I)\left(\beta_{ED} + \epsilon + (A\cos(\phi) + B\sin(\phi))\right), \tag{9}$$

$$\phi_{WU} \approx -\frac{T_{WU}-T_{0x}}{H} + D + \frac{\pi}{2} + \gamma - \tan(I)\left(\beta_{WU} - \epsilon - (A\cos(\phi) + B\sin(\phi))\right), \tag{10}$$

If we consider the zero method, sensor output is zero. The average of Eq. (7)-(10) leads to:

$$\frac{\phi_{EU}+\phi_{WD}+\phi_{ED}+\phi_{WU}}{4} \approx D + \frac{\pi}{2} + \tan(I)\left(\frac{\Sigma\beta_i}{4} + A\cos(\phi) + B\sin(\phi)\right), \tag{11}$$

Only the two initial degrees of freedom and levelling terms remain.

### 2.2 Inclinaison

Inclination development is similar to declination with $\phi = D + k\pi$. The resulting equation (similar to Eq. (11)) is given by:

$$\frac{\beta_{NU}+(\beta_{SD}-\pi)+(2\pi-\beta_{ND})+(\pi-\beta_{NU})}{4} \approx I + (A\cos(D) + B\sin(D)), \tag{12}$$

### 3 Validation

Equations (11)-(12) demonstrate the importance of angular reading accuracy and tilt measurement accuracy in the case of magnetic declination/inclination measurement. Moreover, $\phi$ is related to the True-North while, angle reading is related to the circle index. The way the true north is determined is also critical for declination measurement. Usually, an azimuth mark is pointed. Those three points are investigated here.

### 3.1 Level calibration

Magnetic field is a powerful natural signal that can be used for many purposes. In particular, its stability ensured by the coupling of DIFlux with a variometer provides a nice way to calibrate the AutoDIF tilt sensor. A footscrew is placed in the magnetic meridian while the instrument is setup in inclination measurement, i.e. $\phi = D$ and $T \approx 0$. The "actuated" angle is therefore $(A\cos(D) + B\sin(D))$. The field projection onto the magnetic sensor is given by:

$$T = F\sin(\alpha), \tag{13}$$

Where $\alpha$ is the complementary angle between sensor axis and magnetic field F. When turning the footscrew, the instrument gets tilted in the magnetic meridian. A series of recording allow then to calibrate the level sensor scale factor and its linearity. Magnetic sensor has 0.1nT resolution corresponding to 0.5'' so Fig. 1 does not highlight any linearity error.



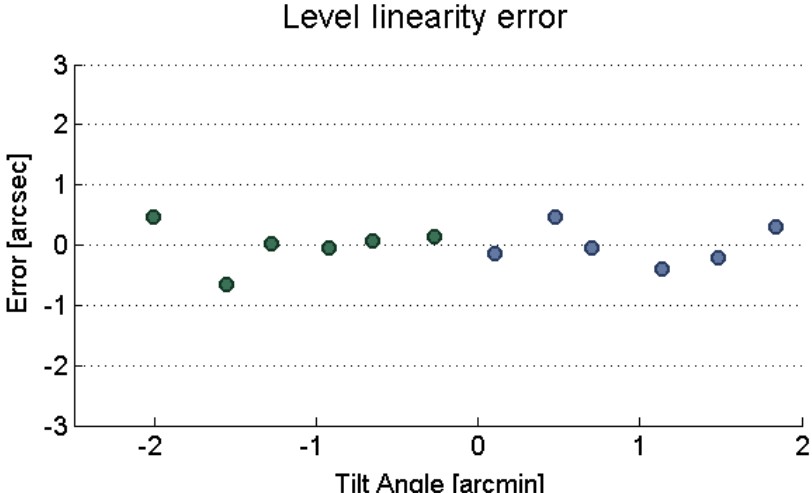

**Figure 1. AutoDIF tilt sensor linearity error. The tilt range is ±2'.**

### 3.2 Angle reading

It is evident that angle reading accuracy in critical when working with a theodolite. Fortunately, ISO 17-123 norm provides a

validation protocol for determining the angle reading uncertainty associated to both horizontal and vertical circles. The principle consists of pointing different directions, then, turning the instrument by 120° and pointing those directions again. Angles between those directions are computed for each set of measurement. Finally, a standard deviation is established. This norm has been directly applied for determining AutoDIF horizontal circle, leading to $1\sigma = 5''$. Comparatively, a Zeiss-010B has been tested and gave $1\sigma = 4''$.

ISO 17-123 is not suitable for validating the vertical circle. First, the norm only check a small part of the circle and then, it requires to point different targets in a vertical plan. Nevertheless, it has been demonstrated above that magnetic sensor in inclination configuration has enough resolution to allow an adaptation of this norm. The inclination measurement set consists of 4 pointing directions driven by the magnetic sensor. By turning this one around the horizontal axis, a new set of 4

positions can be made but now, using another part of the circle. However, angles between each position should be the same. A variometer can be used to take magnetic field variation into account. The standard deviation using this modified method has been determined ($1\sigma = 3''$). Figure 2 shows the circle positions covered by the validation procedure.



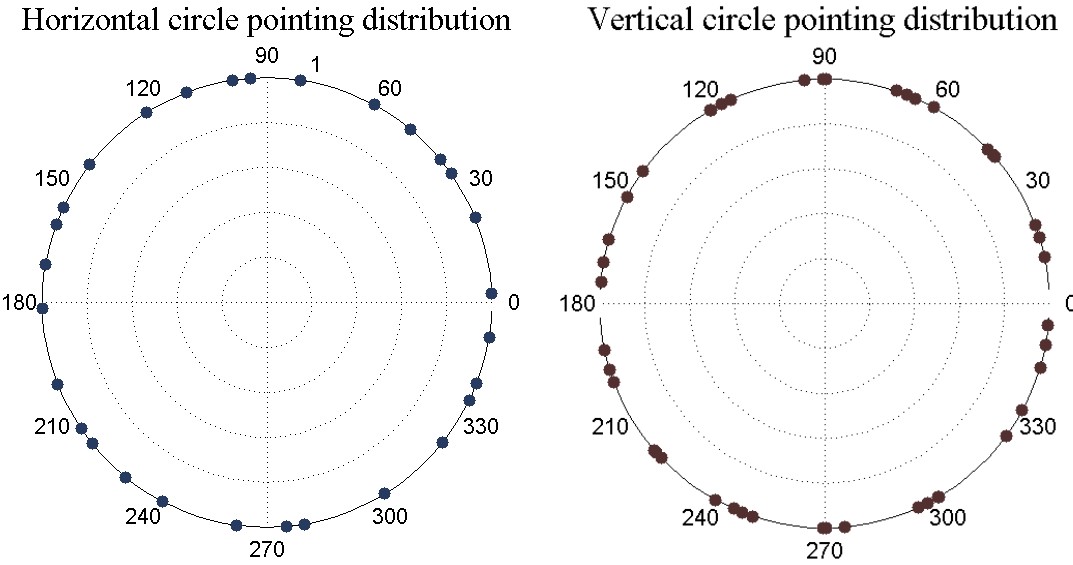

**Figure 2. distribution of pointed direction during ISO-17123 norm validation procedure.**

### 3.3 Azimuth mark

5    The AutoDIF True-North determination system is based on azimuth mark pointing principle. A laser points a retroreflector. The returning beam spot then embedded photocells located on both left and right side of the laser. When the spot is balanced between photocells, difference signal is zero. Figure 3 shows the photocells response while the laser travel the reflector from left to right.

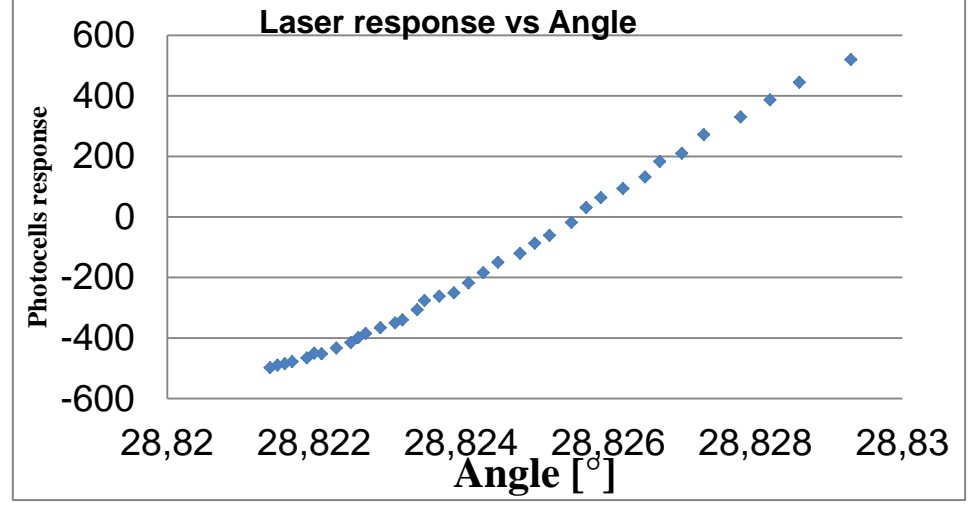

10   **Figure 3. Photocelles response when laser is pointing reflection from left to right side.**





## 4 Experiences

AutoDIF MKII has been presented first during the XIVth IAGA Workshop 2010 in Changchun (China) (Rasson et al. , 2011). Two years later a more reliable version was put in test during the XVth IAGA Workshop in San Fernando (Spain) (Gonsette A. et al. , 2013). This system is now running for about 5 years in Dourbes magnetic observatory.

Experience acquired allowed to identify some weaknesses or possible improvements (Fig.5). For instance, mechanical improvement or better electronic level allowed better magnetic measurement. The software has also evolved giving a real-time computation of the absolute measurement or spot values(declination, inclination and azimuth). A MQTT data transfer protocol has also been implemented for nearly real time data transfer (Bracke, S. et al. , 2016).

We present here a few measurement results. It is more convenient to show instrument baselines rather than vector measurement because this quantity is free of magnetic variation and thus more suitable for DIFlux comparison.

### 4.2 Dourbes

Just after the XVth IAGA workshop, the AutoDIF was installed on the pillar DO2 in the absolute house of Dourbes magnetic

observatory (Fig.4). The azimuth mark is located at about 100m. This pillar is nearly dedicated to this AutoDIF so it can perform absolute measurement every 30 minutes so that we obtained 48 measurements per day (See Fig.4). We just move it twice a week to perform manual absolute measurement for comparison. Because instrument is also dedicated to experiment and improvement tests, big gap are corresponding to immobilization time are presents.

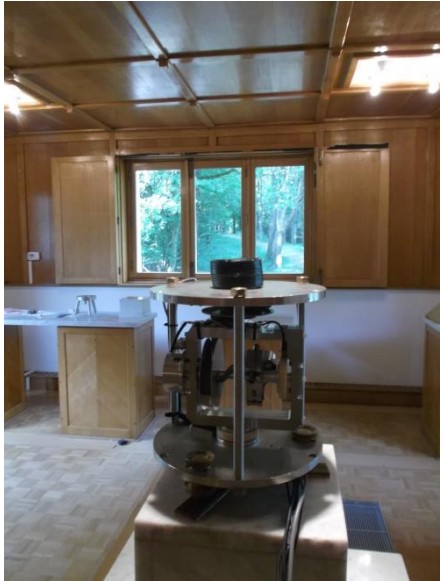

**Figure 4. AutoDIF mk2.2 on pillar DO2 in Dourbes absolute house.**



Figure 4 show comparison baseline of LAMADOU variometer between automatic and manual measurements since beginning of 2016. We can see that for declination, AutoDIF is just over the manual measurement but the variation along the year is the same. The difference is less than 0.005°. For Inclination, the difference is less than 0.002°.

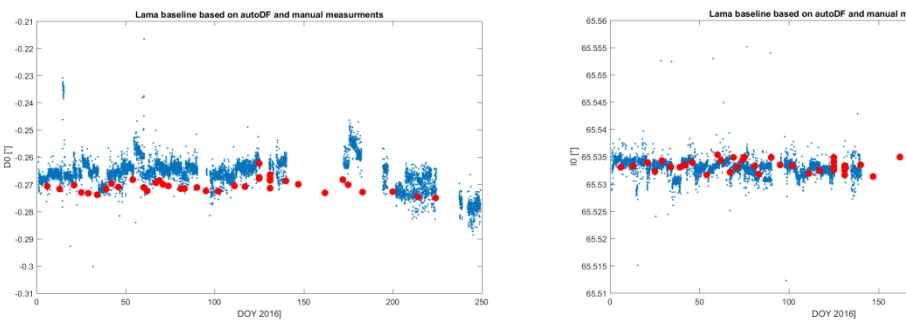

**Figure 5. Intercomparison baselines 2016 for LAMA 2016 of AutoDIF measurement (blue dot) and manual measurement with Zeiss010B**

## 4.3 Conrad

At the end of 2012, we have installed an AutoDIF in the tunnel of the Conrad observatory on the pillar A16 with an azimuth
10   mark at 50m. At the beginning, moist problem did not permit device to work properly (95% relative humidity). This was
solved by heating the instrument in a Plexiglas enclosure (See Fig.6)

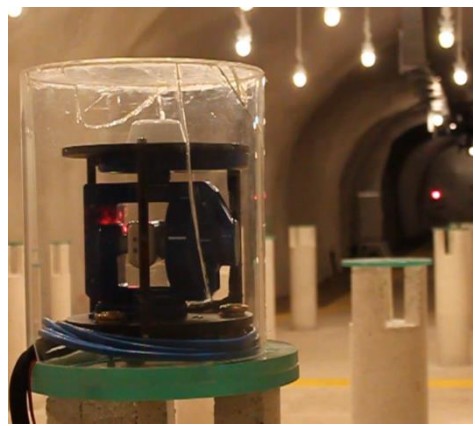

**Figure 6. AutoDIF mk2.2 with his plexiglas cloche on pillar A16 in the tunnel of Conrard observatory.**

In Fig.7, we can see that baselines computed on the AutoDIF measurement are very close from the manual one, except in
15   the last set of data. The gap of automatic data is due to bad contact in the wire of the fluxgate and the laser of the Autodif.
The last dataset are recorded after a quick repair in situ. After that event, the AutoDIF was sent back to Dourbes for repair
and an upgrade.





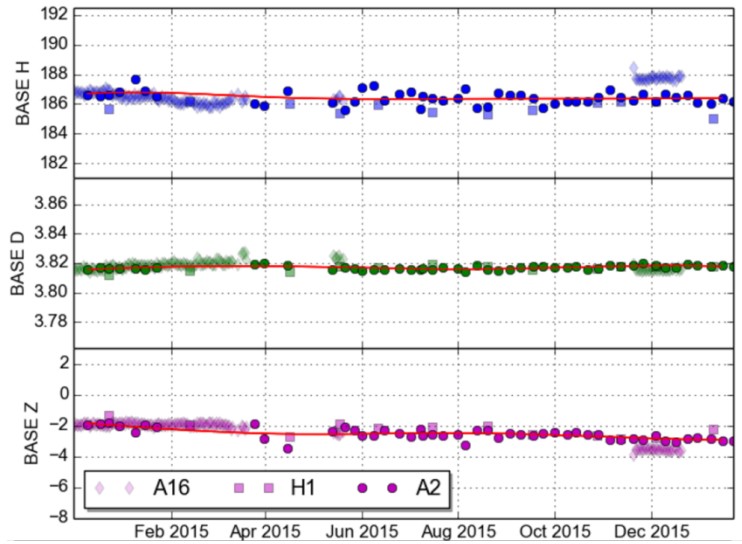

**Figure 7. ABaseline value for FGE variometer , A16 AutoDIF, H1 and A2 manual measurements(Leonhardt, R. et al. , 2015).**

4.4 Choutuppal An AutoDIF participated to intercomparizon session during the XVIthIAGA Workshop in Choutuppal. It performed a measurement set every 30 minutes during one week. Fig.8 shows the baseline results of the inter-comparison session. The gap during night of the first day is due to an unexpected cut of power.

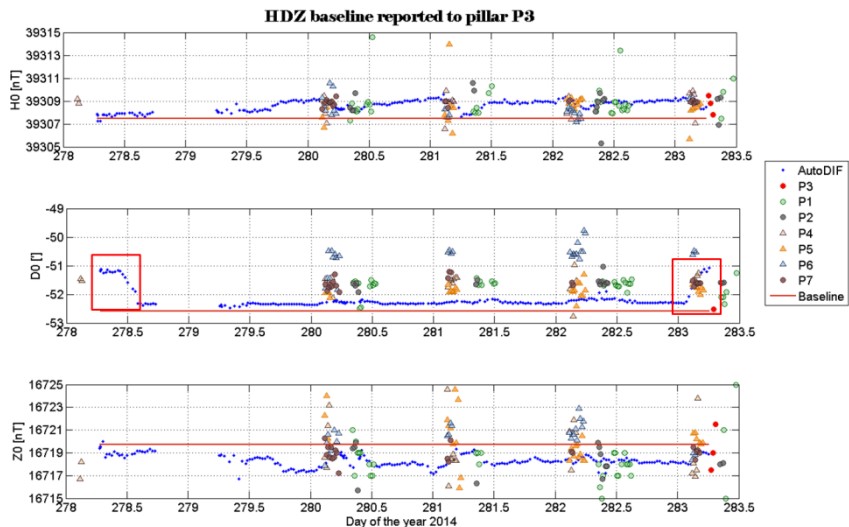

Figure 8. AutoDIF.on his pillar during the 16[th] IAGA Workshop in ChoutuppalThe drop of declination baseline, D0, at beginning comes from a drift of the target which was mounted on a tripod. The Fig.9[left] shows up D0 and down the trace measurement during the week which can explain the D0 variation. We can also observe diurnal variation of the baseline (see Fig.8). That can be explained by the variometer variation (see Fig.9 [right]) probably due to temperature variation in the





variometer room or misalignment of the variometer. This kind of results are only visible when you perform a lot of absolute measurements like with the AutoDIF.

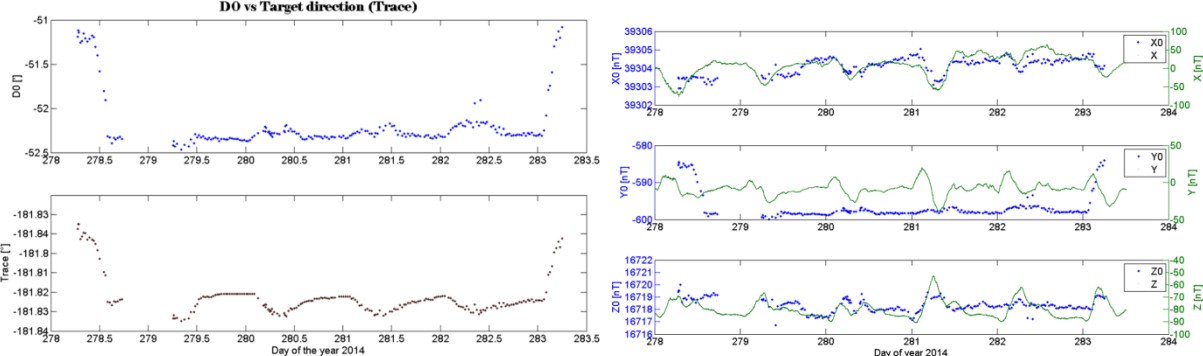

**Figure 9. Left : rift of the target in front of D0 variation. Right : Variometer diurnal variation (green curve) due to temperature variation**

## 5 Conclusions

 After 7 years of development, we can now say that AutoDIF has reached a level of maturity which allows us to envisage its commercialization in observatory. The benefits that they could derive are multiple :

– control manual absolute measurement;

– reduce the number of occurrences of the manual measurement (particularly interesting for unmanned observatory);

– with the possibility of measurement well distributed throughout the day and at high frequency, they can check the correct orientation of their variometer and/or other defects in the installation of these.

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
