# Peer review of "Few years' experience with Automatic DIFlux systems: theory, validation and results."

_Geoscientific Instrumentation, Methods and Data Systems, 2017_

## Referee Comment (RC1) · L. Hegymegi (Referee) · 2 May 2017

Absolute observations in geomagnetic observatories is one of the most important issue since a long time. The presently used method needs experienced operator and it is time consuming. This fact is even more substantial in case of remote stations or hardly accessible places during a certain part of the year. An automatic instrument which helps to overcome on this problem is very important for the observatory community. I strongly support the publication of this article after some additions and corrections.

Chapter 2. Equations (11) and (12) used for declination and inclination angle calculations are correct but in the deduction are some missing things and mistakes (see supplement).

Chapter 3.1 The validation method described here supposes that the magnetometer is

linear in the applied range. Did you tested it?

Usually bubble levels are sensitive to temperature changes. What is your experience in connection with this? You should give some measurement evidence on this problem which can give some information on the environmental requirements of the MKII.3

Chapter 3.2 Your angle reading validation based on ISO 17123 standard. This material is not an open access publication and not known usually for the observatory people. You should give a bit more information about the validation and calculation procedure.

Chapter 3.3 Generally a small size, low-cost laser has not a homogeneous beam. Did you tested how this fact can influence the precision of the azimuth direction determination? Another important question what is the effect of the changing weather conditions on the direction measurement stability? Did you had any long-term test with fixed laser and target pair?

Figure 5. (and not Figure 4. as it is in Chapter 4.2) Exhibits good stability for the baselines in a long period however there are some scatters and deviations in shorter periods (not in the period of contact problems). What was the reason for that? Why the D baseline measured by MKII in most cases is above the baseline measured by manual method?

Chapter 4.3 On Figure 7. Is the red solid line the adopted baseline calculated from manual measurements? What is the dimension of vertical axis?

In Chapter 4.4 the Figure 8. is a good example to show how important is the mutual stability of the instrument pillar (or the instrument itself) and the azimuth mark. This problem is more important when the distance between the instrument and the mirror is smaller. In the paper there is no information about the optimal distance in case of which the reflected laser beam is enough for the differential sensor and the mutual movements of MKII and the laser reflector can be neglected.

What is the reason of the drift at the end of the measurement period? What P1...P7

means?

Chapter 5. What is the reason that from the conclusion is missing that MKII is suitable to run an unmanned observatory? Do you think that this goal was not yet reached?

The paper contains several typing and language errors which does not disturb to understand the article but they should be corrected. For instance the instrument is called sometimes AutoDIF mk2.2 or MarkII Automatic DIFlux etc. IAGA code of Conrad observatory is WIC Name of the relevant ISO standard is ISO 17123 There are cited articles in the main text (line 24) which were not listed in the Reference list etc.

---

## Referee Comment (RC2) · B. Worthington (Referee) · 4 May 2017

Absolute measurements at a Geomagnetic Observatory are critically important for the production of the definitive data product. These measurements are traditionally done by a single person who has been extensively trained. The observations are usually done once or twice a week at observatories where personnel are readily available. As scientific budgets shrink and observatories are established in more remote places, alternatives such as the Automatic DI-Flux are needed. The development of the Automatic DI-Flux is of great interest to the geomagnetic observatory community and this paper provides a valuable update on the progress of this instrument. This paper should be published, however, I do have some comments that should be addressed before publication.

[Figure]

In section 2. A reference is needed for Lauridsen and Kerridge. In the presentation of equation 1, all terms are defined except Ry and Rz. It would make everything clearer if they were defined. In the definitions of the collimation errors, there are references to the "horizontal and vertical plan". I think the correct word would be "plane".

Section 2.1. The equations are simplified by using the small angle approximation. Is this really necessary? There are a lot of scientists in the observatory community who are of the opinion that the small angle approximation should be eliminated in favor of the exact computation.

Section 2.2 Inclination is misspelled in the header. Also, would "Inclination computations are" be better than "Inclination development is".

Section 3.2, first line, the word "in" should be replaced with "is".

Section 3.3, the third sentence would read better if the phrase "strikes the" between the words "then embedded"

Section 4.2. I find figure 5 hard to read, because the graphs are too small. The scales are difficult to read. I would suggest stacking the graphs in a single column so they can be more easily read. The discussion of the results shown in the graph is minimal. A short discussion of the D results would be useful.

Section 4.3. In the line marked 10, I think the word you want is "moisture" instead of "moist". In the following paragraph, where Figure 7, is discussed, you state "except in 10 the last set of data". I think something is missing here. What do you mean by "10"? Is that days, the number of observations, or something else?

Section 4.4. I believe the section title needs to be changed to match the rest of the document. In the last sentence, numbered as 5, I would change the end to "unexpected power failure". It would be helpful if Figure 8 were a little bigger. It is also hard to tell where the figure caption ends and the text resumes. In Figure 9, should the first word of the caption be "Drift"?

Section 5. Is there anything discussed in this paper that could be done to improve or enhance the instrument?

There are a number of minor edits to the English and grammar that would make the paper a little easier to read. I would be happy to help the authors with this. I could send them a scan of my marked up copy if they wish.

---

## Author Comment (AC1) · 7 Jun 2017

First of all thanks for this detailed review with lots of interesting remarks. I only include here some comments on things I didn't include in the new paper and the reasons why.

Chapter 2 Eq. (11) and (12) You are right, I have apply simplification of small angle a bit early. I have corrected it.

Chapter 3.1 Good remark about the fluxgate, we always begin with a test on the linearity of the fluxgate. The probe is set in Noth-Up position. Then, we move the horizontal axis and read the fluxgate response. Like that, we can test scale factor and linearity of the fluxgate. You are true, bubble level are sensitive to temperature, we didn't make experiment on this but we normally install our instrument in absolute house where tem-

perature is nearly still constant. And as we perform turnaround for the measurements, the offset will be compensated and the scale factor . But we will look in to this in a near future in more details. Chapter 3.2 Here is the figure you can find in the norm 17-123

Chapter 3.3 we have no long-term fixed laser but we have an Autodif which perform measurement from a long time with a very good repeatability. When it is foggy the azimuth can not be measured but it is the same for human measurments

Chapter 4.3 the red line is the adopted baseline on the reference pillar

Chapter 4.4 Figure 8 the drift is due to 'play' in the tripod. There is no optimal distance because we can adapt the focus of the laser beam. The maximal distance we use is 150m and the minimal is 50m but in this case we have to be very careful with the stability of the pilar.

Chapter 5 I say that it is particularly interesting to have an AutoDIF in unmanned observatory. But it is always interesting to have a manual measurement when you make maintenance of the instrument.

Please also note the supplement to this comment:
http://www.geosci-instrum-method-data-syst-discuss.net/gi-2017-20/gi-2017-20-AC1-supplement.pdf

⎯⎯⎯⎯⎯⎯⎯⎯⎯⎯⎯⎯⎯⎯⎯⎯

[Figure]

[Figure]

**Fig. 1.** iso17123 figure for H circle

**Supplement:**

**Few years' experience with Automatic DIFlux systems: theory, validation and results.**

Poncelet Antoine[1] , Gonsette Alexandre[1] , and Rasson Jean[1]

[1]Royal Meteorological Institut of Belgium, rue du centre de hysique 4 , 5670 Dourbes

5    Correspondence to: Poncelet (antoine.poncelet@meteo.be)

**Abstract.** The previous release of our Automatic DIFlux, called AutoDIF mk2.2, is now running continuously since June 2012 in the absolute house of Dourbes magnetic observatory performing measurement every 30minutes. A second one is working in the tunnel of Conrad observatory (Austria) since December 2013. After this proof of concept, we improved the

10    AutoDIF to the version mk2.3 which was presented in 16[th] IAGA workshop in Hyderabad. Today, we have successfully deployed 6 AutoDIFs in various environments: 2 in Dourbes (DOU), 1 in Manhay (MAB), 1 in Conrad (CON), 1 in Deajeon (Korea) and 1 is used for tests. This one was installed for 10 month in Chambon-la-Foret (CLF) and since 2016 in Kakioka (KAK). In this paper, we will compare the automatic measurements with the human-made, and discuss the advantages / disadvantages of automatic measurements

**1 Introduction**

After some years of development the release of our Automatic DIFlux (Van Loo et al. , 2007)(Rasson et al., 2009)(Rasson et al. , 2011), called AutoDIF mk2.2, is now running continuously since June 2012 in the absolute house of Dourbes observatory (DOU). A second one is aslo running in Conrad Observatory (WIC) since 2013. These two long time series give us feedback on the strengths and weaknesses of these instruments. This paper will first present the calibration or validation of the most important sensors (fluxgate, level and encoders) and then, the baseline results of AutoDIFmk2.2 for observatories of Dourbes and Conrad. Based on these results and our experience, we will give the improvement we have made to achieve the AutoDIFmk2.3. Then, we will present some baseline results for this last version of AutoDIF.

**2 Theory of Automated DIFlux**

Lauridsen (Lauridsen, 1985) and Kerridge (Kerridge, 1988) had established a DIFlux model giving the magnetic sensor

25    output value according to its orientation in space. On top of the two degrees of freedom (DOF) related to the rotation axes, sensor offset and misalignment errors were taken into account leading to a system with 5 DOF. We propose here to extend those models in order to include a possible levelling error consisting of two angular DOF, one toward geographic north and

the other toward east. The vector magnetic field is then expressed in the sensor frame. The first term of Eq. (1) gives the magnetic sensor output according to its 7 DOF.

$$\vec{T} = \begin{bmatrix} 1 & \gamma & \epsilon \\ -\gamma & 1 & 0 \\ -\epsilon & 0 & 1 \end{bmatrix} R_y(\beta) R_z(\phi) \begin{bmatrix} 1 & 0 & -A \\ 0 & 1 & -B \\ A & B & 1 \end{bmatrix} \begin{bmatrix} X \\ Y \\ Z \end{bmatrix} + \begin{bmatrix} T_{0x} \\ T_{0y} \\ T_{0z} \end{bmatrix},$$
(1)

Where:

- $[X, Y, Z]^t$ — The coordinate system, X pointing North, Y pointing East, Z pointing down
- $A$ — Tilt angle in the geographic north direction
- $B$ — Tilt angle in the east direction
- $T_0$ — Sensor offset
- $\phi$ — Rotation angle around vertical axis
- $\beta$ — Rotation angle around horizontal axis
- $\epsilon$ — Collimation error in a vertical plane
- $\gamma$ — Collimation error in a horizontal plane
- $\vec{T}$ — Magnetic field vector in the sensor frame
- $R_y(\beta)$ — Rotation matrix around horizontal axis of the theodolite
- $R_z(\phi)$ — Rotation matrix around vertical axis of the theodolite

Considering the following transformation: $\begin{bmatrix} X \\ Y \end{bmatrix} = \begin{bmatrix} H \cos(D) \\ H \sin(D) \end{bmatrix}$ the development of the first term of Eq. (1) leads to the AutoDIF model (Eq. 2) .

$T_X =$

$H \cos(\phi - D)(\cos(\beta) - \epsilon \sin(\beta)) - \gamma H \sin(\phi - D) - Z(\sin(\beta) - \epsilon \cos(\beta)) - Z(\cos(\beta) - \epsilon \sin(\beta)(A \cos(\phi) + B \sin(\phi)) + Z\gamma(A\sin(\phi) + B\cos(\phi) - H(\sin(\beta) + \epsilon\cos(\beta))(A \cos(D) + B \sin(D)) + T_{0x},$
(2)

**2.1 Declination**

Declination measurement is performed by putting magnetic sensor perpendicular to the field in a horizontal plane. Therefore, 4 configurations are possible: $\beta = 0, \pi$ and $\phi - D = \frac{\pi}{2}, \frac{3\pi}{2}$ (where $\phi - D$ is the angle of the magnetic azimuth). The 4 configurations are commonly designed according to the sensor pointing direction and its position relative to the horizontal axis. We thus have :

East-UP : $\beta = 0$ and $\phi - D = \frac{\pi}{2}$ ;

West-Down : $\beta = \pi$ and $\phi - D = \frac{\pi}{2}$ ;

East-Down : $\beta = \pi$ and $\phi - D = \frac{3\pi}{2}$ ;

West-UP : $\beta = 0$ and $\phi - D = \frac{3\pi}{2}$.

[revised manuscript text omitted]

**3.3 Azimuth mark**

The AutoDIF True-North determination system is based on azimuth mark pointing principle. A laser points a retroreflector. The returning beam spot strike the embedded photocells located on both left and right side of the laser. When the spot is balanced between photocells, difference signal is zero. Figure 3 shows the photocells response while the laser travel the reflector from left to right.

[Figure]

**Figure 3. Photocelles response when laser is pointing reflection from left to right side.**

**4 Experiences**

AutoDIF MKII has been presented first during the XIVth IAGA Workshop 2010 in Changchun (China) (Rasson et al. ,
2011). Two years later a more reliable version was put in test during the XVth IAGA Workshop in San Fernando (Spain)
(Gonsette A. et al. , 2013). This system is now running for about 5 years in Dourbes magnetic observatory.

Experience acquired allowed to identify some weaknesses or possible improvements (Fig.5). For instance, mechanical
improvement or better electronic level allowed better magnetic measurement. The software has also evolved giving a real-
time computation of the absolute measurement or spot values(declination, inclination and azimuth). A MQTT data transfer
protocol has also been implemented for nearly real time data transfer (Bracke, S. et al. , 2016).

We present here a few measurement results. It is more convenient to show instrument baselines rather than vector
measurement because this quantity is free of magnetic variation and thus more suitable for DIFlux comparison.

**4.2 Dourbes**

Just after the XVth IAGA workshop, the AutoDIF was installed on the pillar DO2 in the absolute house of Dourbes magnetic
observatory (Fig.4). The azimuth mark is located at about 100m. This pillar is nearly dedicated to this AutoDIF so it can
perform absolute measurement every 30 minutes so that we obtained 48 measurements per day (See Fig.5). We just move it
twice a week to perform manual absolute measurement for comparison. Because instrument is also dedicated to experiment
and improvement tests, big gap are corresponding to immobilization time are presents.

[Figure]

**Figure 4. AutoDIF mk2.2 on pillar DO2 in Dourbes absolute house.**

Figure 5 show comparison baseline of LAMADOU variometer between automatic and manual measurements since
5   beginning of 2016. We can see that for declination, AutoDIF is just over the manual measurement but the variation along the
year is the same. The difference is less than 0.005°. For Inclination, the difference is less than 0.002°. These differences may
arise from the instruments which are different (errors on angle readings (see section 3.2)) and/or from target azimuth
determination.

[Figure]

**Figure 5. Intercomparison baselines 2016 for LAMA 2016 of AutoDIF measurement (blue dot) and manual measurement with Zeiss010B**

**5  4.3 Conrad**

At the end of 2012, we have installed an AutoDIF in the tunnel of the Conrad observatory (WIC) on the pillar A16 with an azimuth mark at 50m. At the beginning, moisture problem did not permit device to work properly (95% relative humidity). This was solved by heating the instrument in a Plexiglas enclosure (See Fig.6)

[Figure]

**Figure 6. AutoDIF mk2.2 with his plexiglas cloche on pillar A16 in the tunnel of Conrard observatory (WIC).**

In Fig.7, we can see that baselines computed on the AutoDIF measurement are very close from the manual one, except in the last set of data. The gap of automatic data is due to bad contact in the wire of the fluxgate and the laser of the Autodif. The last dataset are recorded after a quick repair in situ. After that event, the AutoDIF was sent back to Dourbes for repair and an upgrade. The red line is the adopted baseline based on the official pillar A16 where manual measurements are performed.

[Figure]

**Figure 7. Baseline value for FGE variometer , A16 AutoDIF, H1 and A2 manual measurements (H and Z in [nT] and D in [deg]) (Leonhardt, R. et al. , 2015).**

**4.4 Choutuppal**

An AutoDIF participated to intercomparizon session during the XVIthIAGA Workshop in Choutuppal. It performed a measurement set every 30 minutes during one week. Fig.8 shows the baseline results of the inter-comparison session. The gap during night of the first day is due to an unexpected power failure.

[Figure]

**Figure 8. AutoDIF.on his pillar during the 16th IAGA Workshop in Choutuppal**

The drop of declination baseline, D0, at beginning and at the end comes from a drift of the target which was mounted on a tripod. The Fig.9$_{left}$ shows up D0 and down the trace measurement during the week which can explain the D0 variation. This drop can also explain low D0 value for AutoDIF compared with the manual measurements performed on the pillars P1 to P7. This misadventure confirms that both pillars (for the instrument ant for the target azimuth marked) need to be very stable. In this case the azimuth mark was at 100m.  We can also observe diurnal variation of the baseline (see Fig.8). That can be explained by the variometer variation (see Fig.9 $_{right}$) probably due to temperature variation in the variometer room or misalignment of the variometer. These kinds of results are only visible when you perform a lot of absolute measurements like with the AutoDIF.

[Figure]

**Figure 9. Left :Drift of the target in front of D0 variation. Right : Variometer diurnal variation (green curve) due to temperature variation**

**5 Conclusions**

5    After 7 years of development, we can now say that AutoDIF has reached a level of maturity which allows us to envisage its commercialization in observatory. The benefits that they could derive are multiple :

– control manual absolute measurement;

– reduce the number of occurrences of the manual measurement (particularly interesting for unmanned observatory where a manual measurement can be performed every year or 2 year when you visit the observatory);

10    – with the possibility of measurement well distributed throughout the day and at high frequency, they can check the correct orientation of their variometer and/or other defects in the installation of these.

---

## Author Comment (AC2) · 7 Jun 2017

First of all thanks for this detailed review with lots of interesting remarks. I have taken in to account all yours remarks and I only include here some comments on things I didn't include in the new paper and the reasons why.

Section 2.1. The usage of small angles approximations is used for the readability of the equations in the paper. Otherwise, we will have equations which are unreadable. Section 5 It was not the purpose of this article but we are always looking to improve and enhance the instrument.

Please also note the supplement to this comment:

[Figure]

http://www.geosci-instrum-method-data-syst-discuss.net/gi-2017-20/gi-2017-20-AC2-supplement.pdf